# Deep Active Learning with a Neural Architecture Search

**Yonatan Geifman**
Technion – Israel Institute of Technology
`yonatan.g@cs.technion.ac.il`

**Ran El-Yaniv**
Technion – Israel Institute of Technology
`rani@cs.technion.ac.il`

## Abstract

We consider active learning of deep neural networks. Most active learning works in this context have focused on studying effective querying mechanisms and assumed that an appropriate network architecture is a priori known for the problem at hand. We challenge this assumption and propose a novel active strategy whereby the learning algorithm searches for effective architectures on the fly, while actively learning. We apply our strategy using three known querying techniques (softmax response, MC-dropout, and coresets) and show that the proposed approach overwhelmingly outperforms active learning using fixed architectures.

## 1 Introduction

Active learning allows a learning algorithm to control the learning process, by actively selecting the labeled training sample from a large pool of unlabeled instances. Theoretically, active learning has a huge potential, especially in cases where *exponential speedup* in sample complexity can be achieved [10, 25, 9]. Active learning becomes particularly important when considering supervised deep neural models, which are hungry for large and costly labeled training samples. For example, when considering supervised learning of medical diagnoses for radiology images, the labeling of images must be performed by professional radiologists whose availability is scarce and consultation time is costly.

In this paper, we focus on active learning of image classification with deep neural models. There are only a few works on this topic and, for the most part, they concentrate on one issue: How to select the subsequent instances to be queried. They are also mostly based on the *uncertainty sampling* principle in which querying uncertain instances tends to expedite the learning process. For example, [6] employ a Monte-Carlo dropout (MC-dropout) technique for estimating uncertainty of unlabeled instances. [24] applied the well-known softmax response (SR) to estimate uncertainty. [21] and [7] proposed to use coresets on the neural embedding space and then exploit the coreset loss of unlabeled points as a proxy for their uncertainty. A drawback of most of these works is their heavy use of prior knowledge regarding the neural architecture. That is, they utilize an architecture already known to be useful for the classification problem at hand.

When considering active learning of a new learning task, e.g., involving medical images or remote sensing, there is no known off-the-shelf working architecture. Moreover, even if one receives from an oracle the "correct" architecture for the passive learning problem (an architecture that induces the best performance if trained over a very large labeled training sample), it is unlikely that this architecture will be effective in the early stages of an active learning session. The reason is that a large and expressive architecture will tend to overfit when trained over a small sample and, consequently, its generalization performance and the induced querying function (from the overfit model) can be poor (we demonstrate this phenomenon in Section 5).

To overcome this challenge, we propose to perform a *neural architecture search* (NAS) in every active learning round. We present a new algorithm, the *incremental neural architecture search*

(iNAS), which can be integrated together with any active querying strategy. In iNAS, we perform an incremental search for the best architecture from a restricted set of candidate architectures. The motivating intuition is that the capacity of the architectural class should start small, with limited architectural capacity, and should be monotonically non-decreasing along the active learning process. The iNAS algorithm thus only allows for small architectural increments in each active round. We implement iNAS using a flexible architecture family consisting of changeable numbers of stacks, each consisting of a fluid number of Resnet blocks. The resulting active learning algorithm, which we term *active-iNAS*, consistently and significantly improves all known deep active learning algorithms. We demonstrate this advantage of active-iNAS with the above three querying functions over three image classification datasets: CIFAR-10, CIFAR-100, and SVHN.

## 2   Related Work

Active learning has attracted considerable attention since the early days of machine learning. The literature on active learning in the context of classical models such as SVMs is extensive [4, 5, 23, 2, 1, 13], and clearly beyond the scope of this paper. Active learning of deep neural models, as we consider here, has hardly been considered to date. Among the prominent related results, we note Gal et al. [6], who presented active learning algorithms for deep models based on a Bayesian Monte-Carlo dropout (MC-dropout) technique for estimating uncertainty. Wang et al. [24] applied the well-known softmax response (SR) idea supplemented with pseudo-labeling (self-labeling of highly confident points) for active learning. Sener and Savarese [21] and Geifman and El-Yaniv [7] proposed using coresets on the neural embedding space and then exploiting the coreset loss of unlabeled points as a proxy for their uncertainty. A major deficiency of most of these results is that the active learning algorithms were applied with a neural architecture that is already known to work well for the learning problem at hand. This hindsight knowledge is, of course, unavailable in a true active learning setting. To mitigate this problematic aspect, in [7] it was suggested that the active learning be applied only over the "long tail"; namely, to initially utilize a large labeled training sample to optimize the neural architecture, and only then to start the active learning process. This partial remedy suffers from two deficiencies. First, it cannot be implemented in small learning problems where the number of labeled instances is small (e.g., smaller than the "long tail"). Secondly, in Geifman and El-Yaniv's solution, the architecture is fixed after it has been initially optimized. This means that the final model, which may require a larger architecture, is likely to be sub-optimal.

Here, we initiate the discussion of architecture optimization in active learning within the context of deep neural models. Surprisingly, the problem of hyperparameter selection in classical models (such as SVMs) has not been discussed for the most part. One exception is the work of Huang et al. [13] who briefly considered this problem in the context of linear models and showed that active learning performance curves can be significantly enhanced using a proper choice of (fixed) hyperparameters. Huang et al. however, chose the hyperparameters in hindsight. In contrast, we consider a dynamic optimization of neural architectures during the active learning session.

In *neural architecture search* (NAS), the goal is to devise algorithms that automatically optimize the neural architecture for a given problem. Several NAS papers have recently proposed a number of approaches. In [28], a reinforcement learning algorithm was used to optimize the architecture of a neural network. In [29], a genetic algorithm is used to optimize the structure of two types of "blocks" (a combination of neural network layers and building components) that have been used for constructing architectures. The number of blocks comprising the full architecture was manually optimized. It was observed that the optimal number of blocks is mostly dependent on the size of the training set. More efficient optimization techniques were proposed in [16, 19, 20, 18]. In all these works, the architecture search algorithms were focused on optimizing the structure of one (or two) blocks that were manually connected together to span the full architecture. The algorithm proposed in [17] optimizes both the block structure and the number of blocks simultaneously.

## 3   Problem Setting

We first define a standard supervised learning problem. Let $\mathcal{X}$ be a feature space and $\mathcal{Y}$ be a label space. Let $P(X, Y)$ be an unknown underlying distribution, where $X \in \mathcal{X}$, $Y \in \mathcal{Y}$. Based on a labeled training set $S_m = \{(x_i, y_i)\}$ of $m$ labeled training samples, the goal is to select a prediction function $f \in \mathcal{F}$, $f : \mathcal{X} \to \mathcal{Y}$, so as to minimize the risk $R_\ell(f) = \mathbf{E}_{(X,Y)}[\ell(f(x), y)]$, where

$\ell(\cdot) \in \mathbb{R}^+$ is a given loss function. For any labeled set $S$ (training or validation), the empirical risk over $S$ is defined as $\hat{r}_S(f) = \frac{1}{|S|} \sum_{i=1}^{|S|} \ell(f(x_i), y_i)$.

In the pool-based active learning setting, we are given a set $U = \{x_1, x_2, ...x_u\}$ of unlabeled samples. Typically, the acquisition of unlabeled instances is cheap and, therefore, $U$ can be very large. The task of the active learner is to choose points from $U$ to be labeled by an annotator so as to train an accurate model for a given labeling budget (number of labeled points). The points are selected by a query function denoted by $Q$. Query functions often select points based on information inferred from the current model $f_\theta$, the existing training set $S$, and the current pool $U$. In the mini-batch pool-based active learning setting, the $n$ points to be labeled are queried in bundles that are called mini-batches such that a model is trained after each mini-batch.

NAS is formulated as follows. Consider a class $\mathcal{A}$ of architectures, where each architecture $A \in \mathcal{A}$ represents a hypothesis class containing all models $f_\theta \in A$, where $\theta$ represents the parameter vector of the architecture $A$. The objective in NAS is to solve

$$A = \underset{A \in \mathcal{A}}{\operatorname{argmin}} \; \underset{f_\theta \in A|S}{\min} (R_\ell(f)). \tag{1}$$

Since $R_\ell(f)$ depends on an unknown distribution, it is typically proxied by an empirical quantity such as $\hat{r}_S(f)$ where $S$ is a training or validation set.

## 4 Deep Active Learning with a Neural Architecture Search

In this section we define a neural architecture search space over which we apply a novel search algorithm. This search space together with the algorithm constitute a new NAS technique that drives our new active algorithm.

### 4.1 Modular Architecture Search Space

Modern neural network architectures are often modeled as a composition of one or several basic building *blocks* (sometimes referred to as "cells") containing several layers [11, 14, 27, 26, 12]. *Stacks* are composed of several blocks connected together. The full architecture is a sequence of stacks, where usually down-sampling and depth-expansion are performed between stacks. For example, consider the Resnet-18 architecture. This network begins with two initial layers and continues with four consecutive stacks, each consisting of two Resnet *basic blocks*, followed by an average pooling and ending with a softmax layer. The Resnet basic block contains two batch normalized $3{\times}3$ convolutional layers with a ReLU activation and a residual connection. Between every two stacks, the feature maps' resolution is reduced by a factor of 2 (using a strided convolution layer), and the width (number of feature maps in each layer, denoted as $W$) is doubled, starting from 64 in the first block. This classic architecture has several variants, which differ by the number and type of blocks in each stack.

In this work, we consider "homogenous" architectures composed of a single block type and with each stack containing the same number of blocks. We denote such an architecture by $A(B, N_{blocks}, N_{stacks})$, where $B$ is the block, $N_{blocks}$ is the number of blocks in each stack, and $N_{stacks}$ is the number of stacks. For example, using this notation, Resnet-18 is $A(B_r, 2, 4)$ where $B_r$ is the Resnet basic block. Figure 1(a) depicts the proposed homogeneous architecture.

For a given block $B$, we define a modular architecture search space as $\mathcal{A} = \{A(B, i, j) : i \in \{1, 2, 3, ..., N_{blocks}\}, j \in \{1, 2, 3, ..., N_{stacks}\}\}$, which is simply all possible architectures spanned by the grid defined by the two corners $A(B, 1, 1)$ and $A(B, N_{blocks}, N_{stacks})$. Clearly, the space $\mathcal{A}$ is restricted in the sense that it only contains a limited subspace of architectures but nevertheless it contains $N_{blocks} \times N_{stacks}$ architectures with diversity in both numbers of layers and parameters.

### 4.2 Search Space as an Acyclic Directed Graph (DAG)

The main idea in our search strategy is to start from the smallest possible architecture (in the modular search space) and iteratively search for an optimal incremental architectural expansion within the modular search space. We define the *depth* of an architecture to be the number of layers in the architecture. We denote the depth of $A(B, i, j)$ by $|A(B, i, j)| = ij\beta + \alpha$, where $\beta$ is the number of layers

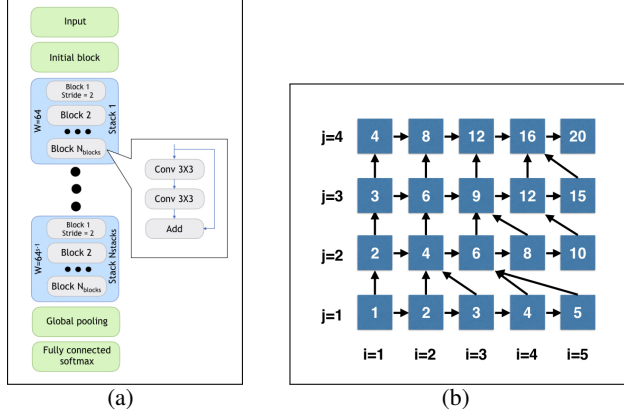

(a)                    (b)

Figure 1: (a) The general proposed architecture contains $N_{blocks}$ blocks in each stack and $N_{stacks}$ stacks. (b) A search space up to $A(B, 5, 4)$ plotted on a grid. The horizontal axis ($i$) represents the number of blocks; the vertical axis ($j$) represents the number of stacks. The arrows represent all the edges of the graph. The number in each vertex is the number of blocks in the architecture ($ij$).

in the block $B$ and $\alpha$ is the number of layers in the *initial block* (all the layers appearing before the first block) plus the number of layers in the *classification block* (all the layers appearing after the last block). It is convenient to represent the architecture search space as a directed acyclic graph (DAG) $G = (V, E)$, where the vertex set $V = \{A(B, i, j)\}$, $B$ is a fixed neural block (e.g., a Resnet basic block), $i \in \{1, 2, \ldots, N_{blocks}\}$ is the number of blocks in each stack, and $j \in \{1, 2, 3, \ldots, N_{stacks}\}$ is the number of stacks. The edge set $E$ is defined based on two incremental expansion steps. The first step, increases the depth of the network without changing the number of stacks (i.e., without affecting the width), and the second step increases the depth while also increasing the number of stacks (i.e., increasing the width). Both increment steps are defined so as to perform the minimum possible architectural expansion (within the search space). Thus, when expanding $A(B, i, j)$ using the first step, the resulting architecture is $A(B, i + 1, j)$. When expanding $A(B, i, j)$ using the second step, we reduce the number of blocks in each stack to perform a minimal expansion resulting in the architecture $A(B, \lfloor \frac{ij}{j+1} \rfloor + 1, j + 1)$. The parameters of the latter architecture are obtained by rounding up the solution $i'$ of the following problem,

$$
\begin{aligned}
i' &= \mathrm{argmin}_{i' > 0} |A(B, i', j + 1)| \\
&s.t. |A(B, i', j + 1)| > |A(B, i, j)|
\end{aligned}
$$

We conclude that each of these steps are indeed depth-expanding. In the first step, the expansion is only made along the depth dimension, while the second step affects the number of stacks and expands the width as well. In both steps, the incremental step is the smallest possible within the modular search space.

In Figure 1(b), we depict the DAG $G$ on a grid whose coordinates are $i$ (blocks) and $j$ (stacks). The modular search space in this example is all the architectures in the range $A(B, 1, 1)$ to $A(B, 5, 4)$. The arrows represents all edges in $G$. In this formulation, it is evident that every path starting from any architecture can be expanded up to the largest possible architecture. Moreover, every architecture is reachable when starting from the smallest architecture $A(B, 1, 1)$. These two properties serve well our search strategy.

### 4.3 Incremental Neural Architecture Search

The proposed *incremental neural architecture search* (iNAS) procedure is described in Algorithm 1 and operates as follows. Given a small initial architecture $A(B, i_0, j_0)$, a training set $S$, and an architecture search space $\mathcal{A}$, we first randomly partition the set $S$ into training and validation subsets, $S'$ and $V'$, respectively, $S = S' \cup V'$. On iteration $t$, a set of candidate architectures is selected based on the edges of the search DAG (see Section 4.2) including the current architecture and the two connected vertices (lines 5-6). This step creates a candidate set, $\mathcal{A}'$, consisting of three models, $\mathcal{A}' = \{A(B, i, j), A(B, \lfloor \frac{ij}{j+1} \rfloor + 1, j + 1), A(B, i + 1, j)\}$. In line 7, the best candidate in terms

of validation performance is selected and denoted $A_t = A(B, i_t, j_t)$. The optimization problem formulated in line 7 is an approximation of the NAS objective formulated in Equation (1). The algorithm terminates whenever $A_t = A_{t-1}$, or a predefined maximum number of iterations is reached (in which case $A_t$ is the final output).

---

**Algorithm 1** iNAS

1: **iNAS**$(S, A(B, i_0, j_0), \mathcal{A}, T_{iNAS})$
2: Let $S', V'$ be an train-test random split of $S$
3: **for** t=1:$T_{iNAS}$ **do**:
4:     $i \leftarrow i_{t-1}; j \leftarrow j_{t-1}$
       $\mathcal{A}' = \{ \quad A(B, i, j),$
5:         $A(B, \lfloor \frac{ij}{j+1} \rfloor + 1, j + 1),$
       $A(B, i + 1, j) \}$
6:     $\mathcal{A}' = \mathcal{A}' \cap \mathcal{A}$
       $A(B, i_t, j_t) =$
7:     $= \operatorname{argmin}_{A \in \mathcal{A}'} \hat{r}_{V'}(\operatorname{argmin}_{f_\theta \in A} \hat{r}_{S'}(f_\theta))$
8:     **if** $A(B, i_t, j_t) = A(B, i_{t-1}, j_{t-1})$ **then**
9:         break
10:    **end if**
11: **end for**
12: Return $A(B, i_t, j_t)$

---

**Algorithm 2** Deep Active Learning with iNAS

1: **active-iNAS**$(U, A_0, \mathcal{A}, Q, b, k)$
2: $t \leftarrow 1$
3: $S_t \leftarrow$ **Sample $k$ points from $U$ at random**
4: $U_0 \leftarrow U \backslash S_1$
5: **while true do**
6:     $A_t \leftarrow$ **iNAS**$(S, A_{t-1}, \mathcal{A})$
7:     **train** $f_\theta \in A_t$ **using** $S$
8:     **if budget exhausted or** $U_t = \emptyset$ **then**
9:         **Return** $f_\theta$
10:    **end if**
11:    $S' \leftarrow Q(f_\theta, S_t, U_t, b)$
12:    $S_{t+1} \leftarrow S_t \cup S'$
13:    $U_{t+1} \leftarrow U_t \backslash S'$
14:    $t \leftarrow t + 1$
15: **end while**

---

### 4.4 Active Learning with iNAS

The *deep active learning with incremental neural architecture search* (active-iNAS) technique is described in Algorithm 2 and works as follows. Given a pool $U$ of unlabeled instances from $\mathcal{X}$, a set of architectures $\mathcal{A}$ is induced using a composition of basic building blocks $B$ as shown in Section 4.1, an initial (small) architecture $A_0 \in \mathcal{A}$, a query function $Q$, an initial (passively) labeled training set size $k$, and an active learning batch size $b$. We first sample uniformly at random $k$ points from $U$ to constitute the initial training set $S_1$. We then iterate the following three steps. First, we search for an optimal neural architecture using the iNAS algorithm over the search space $\mathcal{A}$ with the current training set $S_t$ (line 6). The initial architecture for iNAS is chosen to be the selected architecture from the previous active round ($A_{t-1}$), assuming that the architecture size is non-decreasing along the active learning process. The resulting architecture at iteration $t$ is denoted $A_t$. Next, we train a model $f_\theta \in A_t$ based on $S_t$ (line 7). Finally, if the querying budget allows, the algorithm requests $b$ new points using $Q(f_\theta, S_t, U_t, b)$ and updates $S_{t+1}$ and $U_{t+1}$ correspondingly. Otherwise the algorithm returns $f_\theta$ (lines 8-14).

### 4.5 Theoretical Motivation and Implementation Notes

The iNAS algorithm is designed to exploit the prior knowledge gleaned from samples of increasing size, which is motivated from straightforward statistical learning arguments. iNAS starts with small capacity so as to avoid over-fitting in early stages, and then allows for capacity increments as labeled data accumulates. Recall from statistical learning theory that for a given hypothesis class $\mathcal{F}$ and training set $S_m$, the generalization gap can be bounded as follows with probability at least $1 - \delta$,

$$R(f) - \hat{r}_{S_m}(f) \leq O(\sqrt{\frac{d \log(m/d) + \log(1/\delta)}{m}}),$$

where $d$ is the VC-dimension of $\mathcal{F}$. Recently, Bartlett et al. [3] showed a nearly tight bound for the VC-dimension of deep neural networks. Let $W$ be the number of parameters in a neural network, let $L$ be the number of layers, and $U$, the number of computation units (neurons/filters), [3] showed that the VC dimension of ReLU-activated regression models is bounded as $VCdim(\mathcal{F}) \leq O(\bar{L}W \log(U))$, where $\bar{L} \triangleq \frac{1}{W} \sum_{i=1}^{L} W_i$ and $W_i$ is the number of parameters from the input to layer $i$. As can be seen, the expansion steps proposed in iNAS are designed to minimally expand the VC-dimension of $\mathcal{F}$. When adding blocks, $W$, $U$ and $\bar{L}$ grow linearly. As a result, the VC-dimension grows linearly. When adding a stack (in the iNAS algorithm), $W$ and $U$ grow sub-exponentially, and $L$ (and $\bar{L}$) also grows. Along the active learning session, $m$ grows linearly in incremental steps,

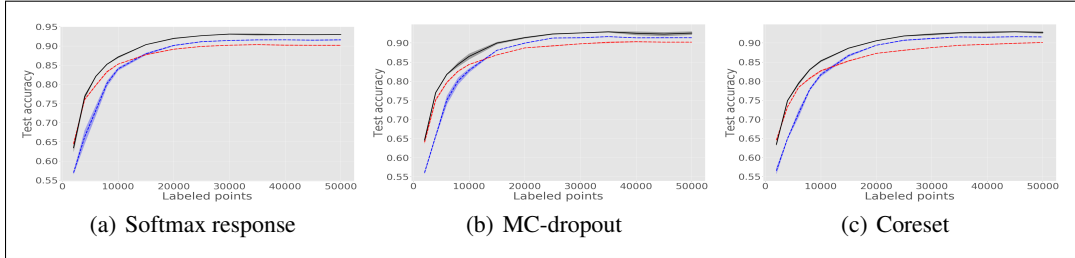

Figure 2: Active learning curves for CIFAR-10 dataset using various query functions, (a) softmax response, (b) MC-dopout, (c) coreset. In black (solid) – Active-iNAS (ours), blue (dashed) – Resnet-18 fixed architecture, and red (dashed) – $A(B_r, 1, 2)$ fixed.

thus, it motivates a linear growth in the VC-dimension (in incremental steps) so as to maintain the generalization gap bound as small as possible. Alternate approaches that are often used, such as a full grid-search on each active round, would not enjoy these benefits and will be prone to overfitting (not to mention that full-grid search could be computationally prohibitive).

Turning now to analyze the run time of active-iNAS, when running with small active learning mini-batches, it is evident that the iNAS algorithm will only require one iteration at each round, resulting in only having to train three additional models at each round. In our implementation of iNAS, we apply "premature evaluation" as considered in [22]; our models are evaluated after $T_{SGD}/4$ epochs where $T_{SGD}$ is the total number of epochs in each round. Our final active-iNAS implementation thus only takes $1.75T_{SGD}$ for each active round. For example, in the CIFAR-10 experiment $T_{SGD} = 200$ requires less than 2 GPU hours (on average) for an active learning round (Nvidia Titan-Xp GPU).

## 5    Experiments

We first compare active-iNAS to active learning performed with a fixed architecture over three datasets, we apply three querying functions, softmax response, coresets and MC-dropout. Then we analyze the architectures learned by iNAS along the active process. We also empirically motivate the use of iNAS by showing how optimized architecture can improve the query function. Finally, we compare the resulting active learning algorithm obtained with the active-iNAS framework.

### 5.1    Experimental Setting

We used an architecture search space that is based on the Resnet architecture [11]. The *initial block* contains a convolutional layer with filter size of $3 \times 3$ and depth of 64, followed by a max-pooling layer having a spatial size of $3 \times 3$ and strides of 2. The *basic block* contains two convolutional layers of size $3 \times 3$ followed by a ReLU activation. A residual connection is added before the activation of the second convolutional layer, and a batch normalization [15] is used after each layer. The *classification block* contains an average pooling layer that reduces the spatial dimension to $1 \times 1$, and a fully connected classification layer followed by softmax. The search space is defined according to the formulation in Section 4.1, and spans all architectures in the range $A(B_r, 1, 1)$ to $A(B_r, 12, 5)$.

As a baseline, we chose two fixed architectures. The first architecture was the one optimized for the first active round (optimized over the initial seed of labeled points), and which coincidentally happened to be $A(B_r, 1, 2)$ on all tested datasets. The second architecture was the well-known Resnet-18, denoted as $A(B_r, 2, 4)$, which is some middle point in our search grid.

We trained all models using *stochastic gradient descent* (SGD) with a batch size of 128 and momentum of 0.9 for 200 epochs. We used a learning rate of 0.1, with a learning rate multiplicative decay of 0.1 after epochs 100 and 150. Since we were dealing with different sizes of training sets along the active learning process, the epoch size kept changing. We fixed the size of an epoch to be 50,000 instances (by oversampling), regardless of the current size of the training set $S_t$. A weight decay of $5e$-4 was used, and standard data augmentation was applied containing horizontal flips, four pixel shifts and up to 15-degree rotations.

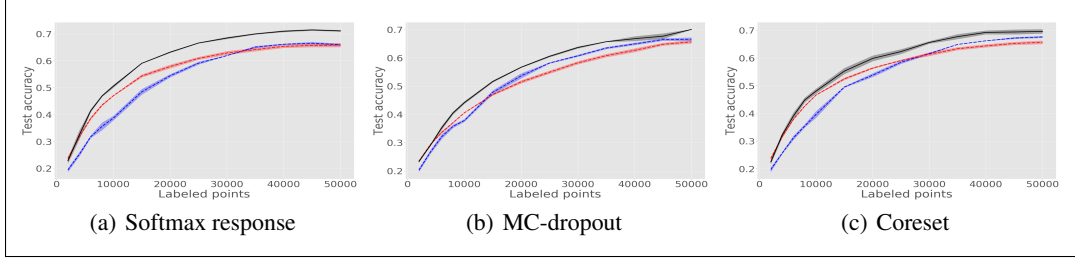

Figure 3: Active learning curves for CIFAR-100 dataset using various query functions, (a) softmax response, (b) MC-dopout, (c) coreset. In black (solid) – Active-iNAS (ours), blue (dashed) – Resnet-18 fixed architecture, and red (dashed) – $A(B_r, 1, 2)$ fixed.

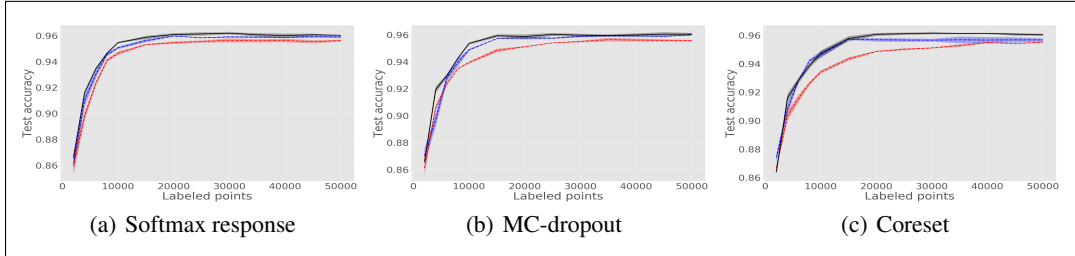

Figure 4: Active learning curves for SVHN dataset using various query functions, (a) softmax response, (b) MC-dopout, (c) coreset. In black (solid) – Active-iNAS (ours), blue (dashed) – Resnet-18 fixed architecture, and red (dashed) – $A(B_r, 1, 2)$ fixed.

The active learning was implemented with an initial labeled training seed ($k$) of 2000 instances. The active mini-batch size ($b$) was initialized to 2000 instances and updated to 5000 after reaching 10000 labeled instances. The maximal budget was set to 50,000 for all datasets[1]. For time efficiency reasons, the iNAS algorithm was implemented with $T_{iNAS} = 1$, and the training of new architectures in iNAS was early-stopped after 50 epochs, similar to what was done in [22].

## 5.2 Active-iNAS vs. Fixed Architecture

The results of an active learning algorithm are often depicted by a curve measuring the trade-off between labeled points (or a budget) vs. performance (accuracy in our case). For example, in Figure 2(a) we see the results obtained by active-iNAS and two fixed architectures for classifying CIFAR-10 images using the softmax response querying function. In black (solid), we see the curve for the active-iNAS method. The results of $A(B_r, 1, 2)$ and Resnet-18 ($A(B_r, 2, 4)$) appear in (dashed) red and (dashed) blue, respectively. The $X$ axis corresponds to the labeled points consumed, starting from $k = 2000$ (the initial seed size), and ending with 50,000 . In each active learning curve, the standard error of the mean over three random repetitions is shadowed.

We present results for CIFAR-10, CIFAR-100 and SVHN. We first analyze the results for CIFAR-10 (Figure 2). Consider the graphs corresponding to the fixed architectures (red and blue). It is evident that for all query functions, the small architecture (red) outperforms the big one (Resnet-18 in blue) in the early stage of the active process. Later on, we see that the big and expressive Resnet-18 outperforms the small architecture. Active-iNAS, performance consistently and significantly outperforms both fixed architectures almost throughout the entire range. It is most striking that active-iNAS is better than each of the fixed architectures even when all are consuming the entire training budget. Later on we speculate about the reason for this phenomenon as well as the switch between the red and blue curves occurring roughly around 15,000 training points (in Figure 2(a)).

Turning now to CIFAR-100 (Figure 3), we see qualitatively very similar behaviors and relations between the various active learners. We now see that the learning problem is considerably harder, as indicated by the smaller area under all the curves. Nevertheless, in this problem active-iNAS

achieves a substantially greater advantage over the fixed architectures in all three query functions. Finally, in the SVHN digit classification task, which is known to be easier than both the CIFAR tasks, we again see qualitatively similar behaviors that are now much less pronounced, as all active learners are quite effective. On the other hand, in the SVHN task, active-iNAS impressively obtains almost maximal performance after consuming only 20% of the training budget.

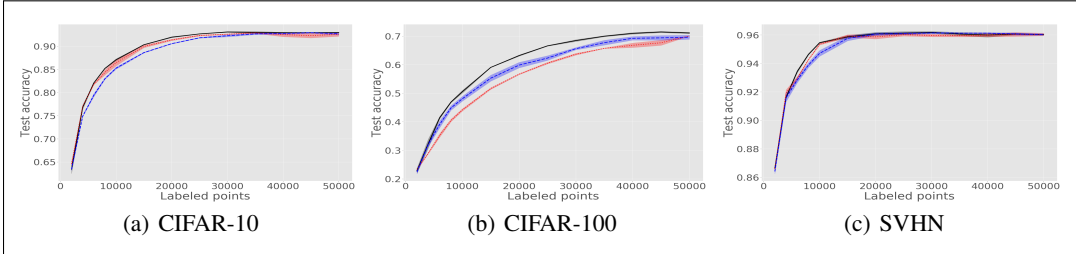

(a) CIFAR-10          (b) CIFAR-100          (c) SVHN

Figure 5: Comparison of active-iNAS for various query functions across three datasets, (a) CIFAR10, (b) CIFAR-100, (c) SVHN. In black (solid) – softmax response, red (dashed) – MC-dropout, and blue (dashed) – coreset.

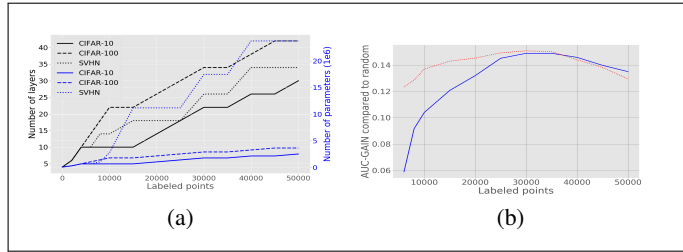

(a)          (b)

Figure 6: (a) The architectures learned by iNAS as function of the labeled samples. Blue curves represent number of parameters ($\times 1e6$) and black curves represent the number of layers. CIFAR-10 – solid line, CIFAR-100 – dashed line and SVHN – dotted line. (b) Comparison of AUC-GAIN for softmax response active learning over CIFAR-10 for two architectures. In red (solid), the small architecture ($A(B_r, 1, 2)$) and in blue (dashed) the Resnet-18 architecture ($A(B_r, 2, 4)$).

## 5.3 Analyzing the Learned Architectures

In addition to standard performance results presented in Section 5.2, it is interesting to inspect the sequence of architectures that have been selected by iNas along the active learning process. In Figure 6(a) we depict this dynamics; for example, consider the CIFAR-10 dataset appearing in solid lines, where the blue curve represents the number of parameters in the network and the black shows the number of layers in the architecture. Comparing CIFAR-10 (solid) and CIFAR-100 (dashed), we see that active-iNAS prefers, for CIFAR-100, deeper architectures compared to its choices for CIFAR-10. In contrast, in SVHN (dotted), active-iNAS gravitates to shallower but wider architectures, which result significantly larger numbers of parameters. The iNAS algorithm is relatively stable in the sense that in the vast majority of random repeats of the experiments, similar sequences of architectures have been learned (this result is not shown in the figure).

A hypothesis that might explain the latter results is that CIFAR-100 contains a larger number of "concepts" requiring deeper hierarchy of learned CNN layers compared to CIFAR-10. The SVHN is a simpler and less noisy learning problem, and, therefore, larger architectures can play without significant risk of overfitting.

### 5.4 Enhanced Querying with Active-iNAS

In this section we argue and demonstrate that optimized architectures not only improve generalization at each step, but also enhance the query function quality[2]. In order to isolate the contribution of the query function, we normalize the active performance by the performance of a passive learner obtained with the same model. A common approach for this normalization has already been proposed in [13, 2], we define conceptually similar normalization as follows. Let the *relative AUC gain* be the relative reduction of area under the curve (AUC) of the 0-1 loss in the active learning curve, compared to the AUC of the passive learner (trained over the same number of random queries, at each round); namely, AUC-GAIN$(PA, AC, m) = \frac{AUC_m(PA) - AUC_m(AC)}{AUC_m(PA)}$, where $AC$ is an active learning algorithm, $PA$ is its passive application (with the same architecture), $m$ is a labeling budget, and $AUC_m(\cdot)$ is the area under the learning curve (0-1 loss) of the algorithm with budget $m$. Clearly, high values of AUC-GAIN correspond to high performance and vice versa.

In Figure 6(b), we used the AUC-GAIN to measure the performance of the softmax response querying function on the CIFAR-10 dataset over all training budgets up to the maximal (50,000). We compare the performance of this query function applied over two different architectures: the small architecture ($A(B_r, 1, 2)$), and Resnet-18 ($A(B_r, 2, 4)$. We note that it is unclear how to define AUC-GAIN for active-iNAS because it has a dynamically changing architecture.

As can easily be seen, the small architecture dramatically outperforms Resnet-18 in the early stages. Later on, the AUC-GAIN curves switch, and Resnet-18 catches up and outperforms the small architecture. This result supports the intuition that improvements in the generalization tend to improve the effectiveness of the querying function. We hypothesize that the active-iNAS' outstanding results shown in Section 5.2 have been achieved not only by the improved generalization of every single model, but also by the effect of the optimized architecture on the querying function.

### 5.5 Query Function Comparison

In Section 5.2 We demonstrated that active-iNAS consistently outperformed direct active applications of three querying functions. Here, we compare the performance of the three active-iNAS methods, applied with those three functions: softmax response, MC-dropout and coreset. In Figure 6 we compare these three active-iNAS algorithms over the three datasets. In all three datasets, softmax response is among the top performers, whereas one or the other two querying functions is sometimes the worst. Thus, softmax response achieves the best results. For example, on CIFAR-10 and SVHN, the MC-dropout is on par with softmax, but on CIFAR-100 MC-dropout is the worst.

The poor performance of MC-dropout over CIFAR-100 may be caused by the large number of classes, as pointed out by [8] in the context of selective classification. In all cases, coreset is slightly behind the softmax response. This is in sharp contrast to the results presented by [21] and [7]. We conclude this section by emphasizing that our results indicate that the combination of softmax response with active-iNAS is the best active learning method.

## 6 Concluding Remarks

We presented active-iNAS, an algorithm that effectively integrates deep neural architecture optimization with active learning. The active algorithm performs a monotone search for the locally best architecture on the fly. Our experiments indicate that active-iNAS outperforms standard active learners that utilize suitable and commonly used fixed architecture. In terms of absolute performance quality, to the best of our knowledge, the combination of active-iNAS and softmax response is the best active learner over the datasets we considered.

## Acknowledgments

This research was supported by The Israel Science Foundation (grant No. 81/017).

## Footnotes

[1]SVHN contains 73,256 instances and was, therefore, trimmed to 50000.

[2]We only consider querying functions that are defined in terms of a model (such as all query functions considered here).

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
