[Supplementary Material · Active_Learning_with_Neural_Architecture_Search (9).pdf]

# Deep Active Learning with a Neural Architecture Search - Supplementary Material

## 1 Related Work

Active learning has attracted considerable attention since the early days of machine learning. The literature on active learning in the context of classical models such as SVMs is extensive [3, 6, 22, 2, 1, 11], and clearly beyond the scope of this paper. Active learning of deep neural models, as we consider here, has hardly been considered to date. Among the prominent related results, we note Gal et al. [8], who presented active learning algorithms for deep models based on a Bayesian Monte-Carlo dropout (MC-dropout) technique for estimating uncertainty. Wang et al. [23] applied the well-known softmax response (SR) idea supplemented with pseudo-labeling (self-labeling of highly confident points) for active learning. Sener and Savarese [20] and Geifman and El-Yaniv [9] proposed using coresets on the neural embedding space and then exploiting the coreset loss of unlabeled points as a proxy for their uncertainty. A major deficiency of most of these results is that the active learning algorithms were applied with a neural architecture that is already known to work well for the learning problem at hand. This hindsight knowledge is, of course, unavailable in a true active learning setting. To mitigate this problematic aspect, in [9] it was suggested that the active learning be applied only over the "long tail"; namely, to initially utilize a large labeled training sample to optimize the neural architecture, and only then to start the active learning process. This partial remedy suffers from two deficiencies. First, it cannot be implemented in small learning problems where the number of labeled instances is small (e.g., smaller than the "long tail"). Secondly, in Geifman and El-Yaniv's solution, the architecture is fixed after it has been initially optimized. This means that the final model, which may require a larger architecture, is likely to be sub-optimal.

Here, we initiate the discussion of architecture optimization in active learning within the context of deep neural models. Surprisingly, the problem of hyperparameter selection in classical models (such as SVMs) has not been discussed for the most part. One exception is the work of Huang et al. [11] who briefly considered this problem in the context of linear models and showed that active learning performance curves can be significantly enhanced using a proper choice of (fixed) hyperparameters. Huang et al. however, chose the hyperparameters in hindsight. In contrast, we consider a dynamic optimization of neural architectures during the active learning session.

In *neural architecture search* (NAS), the goal is to devise algorithms that automatically optimize the neural architecture for a given problem. Several NAS papers have recently proposed a number of approaches. In [24], a reinforcement learning algorithm was used to optimize the architecture of a neural network. In [25], a genetic algorithm is used to optimize the structure of two types of "blocks" (a combination of neural network layers and building components) that have been used for constructing architectures. The number of blocks comprising the full architecture was manually optimized. It was observed that the optimal number of blocks is mostly dependent on the size of the training set. More efficient optimization techniques were proposed in [14, 18, 19, 16]. In all these works, the architecture search algorithms were focused on optimizing the structure of one (or two) blocks that were manually connected together to span the full architecture. The algorithm proposed in [15] optimizes both the block structure and the number of blocks simultaneously.

When considering NAS for fully-connected networks, [4] proposed an algorithm that iteratively adds neurons to an existing layer or to initiate a new layer. Their algorithm iteratively optimizes the width and depth of a network. For a comprehensive survey on NAS techniques, see [5]. To the best of our knowledge, no work has been done on architecture searches for active learning.

## 2 Experimental Design and Details

### 2.1 Datasets

**CIFAR-10.** The CIFAR-10 dataset [13] is an image classification dataset containing 50,000 training images and 10,000 test images that are classified into 10 categories. The image size is $32 \times 32 \times 3$ pixels (RGB images).

**CIFAR-100.** The CIFAR-100 dataset [13] is an image classification dataset containing 50,000 training images and 10,000 test images that are classified into 100 categories. The image size is $32 \times 32 \times 3$ pixels (RGB images).

**Street View House Numbers (SVHN).** The SVHN dataset [17] is an image classification dataset containing 73,257 training images and 26,032 test images classified into 10 classes representing digits. The images are digits of house numbers cropped and aligned, taken from the Google Street View service. Image size is $32 \times 32 \times 3$ pixels.

### 2.2 Architectures and Hyperparameters

We used an architecture search space that is based on the Resnet architecture [10]. The *initial block* contains a convolutional layer with filter size of $3 \times 3$ and depth of 64, followed by a max-pooling layer having a spatial size of $3 \times 3$ and strides of 2. The *basic block* contains two convolutional layers of size $3 \times 3$ followed by a ReLU activation. A residual connection is added before the activation of the second convolutional layer, and a batch normalization [12] is used after each layer. The classification block contains an average pooling layer that reduces the spatial dimension to $1 \times 1$, and a fully connected classification layer followed by softmax. The search space is defined according to the formulation in Section 3, and spans all architectures in the range $A(B_r, 1, 1)$ to $A(B_r, 12, 5)$.

As a baseline, we chose two fixed architectures. The first architecture was the one optimized for the first active round (optimized over the initial seed of labeled points), and which coincidentally happened to be $A(B_r, 1, 2)$ on all tested datasets. The second architecture was the well-known Resnet-18, denoted as $A(B_r, 2, 4)$, which is some middle point in our search grid.

We trained all models using *stochastic gradient descent* (SGD) with a batch size of 128 and momentum of 0.9 for 200 epochs. We used a learning rate of 0.1, with a learning rate multiplicative decay of 0.1 after epochs 100 and 150. Since we were dealing with different sizes of training sets along the active learning process, the epoch size kept changing. We fixed the size of an epoch to be 50,000 instances (by oversampling), regardless of the current size of the training set $S_t$. A weight decay of $5e\text{-}4$ was used, and standard data augmentation was applied containing horizontal flips, four pixel shifts and up to 15-degree rotations.

The active learning was implemented with an initial labeled training seed ($k$) of 2000 instances. The active mini-batch size ($b$) was initialized to 2000 instances and updated to 5000 after reaching 10000 labeled instances. The maximal budget was set to 50,000 for all datasets[1]. For time efficiency reasons, the iNAS algorithm was implemented with $T_{iNAS} = 1$, and the training of new architectures in iNAS was early-stopped after 50 epochs, similar to what was done in [21].

### 2.3 Query Functions

We applied the following three well known query functions.

**Softmax Response.** The softmax response method (SR) simply estimates prediction confidence (the inverse of uncertainty) by the maximum softmax value of the instance. In the batch pool-based active learning setting that we consider here, we simply query labels for the least confident $b$ points.

Figure 1: Comparison of active-iNAS to coreset of Sener and Savarese [20] across two datasets, (a) CIFAR-10, (b) CIFAR-100. In black (solid) – active-iNAS, red (dashed) – coreset (Sener and Savarese).

**MC-dropout [8, 7].** The points in the pool are ordered based on their prediction uncertainty estimated by MC-dropout and queried in that order. The MC-dropout was implemented with $p = 0.5$ for the dropout rate and 100 feed-forward iterations for each sample.

**Coreset [20, 9].** The coreset method was implemented as follows. For a trained model $f$, we denote the output of the representation layer (the layer before the last) as $\phi_f$. For every sample $x \in U$, its coreset loss is measured as $\min_{x' \in S}(d(\phi_f(x'), \phi_f(x))$, where $d$ is the $l2$ euclidean distance. We iteratively sampled the point with the highest coreset loss with respect to the latest set $S$, $b$ times.

## 2.4 Code for Active-iNAS

The code for all the experiments performed in the paper can be found in the following (anonymized) link. https://github.com/anonygit32/active_inas

## 3 Direct Comparison to Other Work

We compare performance of active-iNAS to two other deep active learning papers. We begin by considering the algorithm proposed by Sener and Savarese [20] in the the last ICLR . To compare apples to apples we use the setting proposed by these authors and apply their active training schedule consisting of five active rounds applied with 5,000, 10,000, 15,000, 20,000, and 50,000 labeled points; see [20] for details. Their algorithm is based on the coreset query function (that has also been discussed in our paper) to train a deep model with the VGG-16 architecture whose selection considerations were not discussed. In our comparison, we use the results reported in [20] for the CIFAR-10 and CIFAR-100 datasets. The learning curves in Figure 1 show their reported results (dashed, red) and ours (solid, black) applied with the softmax response. Obviously, active-iNAS is a clear winner. It is clearly evident that the active-iNAS outperforms the coreset of [20] on all budgets.

Motivated by similar prior-knowledge considerations as discussed here, Geifman and El-Yaniv [9] proposed the "long tail" active learning setting in which a (relatively large) labeled set is initially queried at random (i.e., passively) and used for acquiring "prior knowledge" (e.g., to select an architecture). As Sener and Savarese, these authors also proposed to utilize the the coreset idea to form a querying function. Here we compare their results to our approach, in which active learning is operated from the start of the active session with iNAS. Is it the case that a large labeled training set can be used to acquire sufficient knowledge to select a useful architecture? Can our approach compete with such knowledge?

The results are presented in Figure 2 showing active-iNAS applied with softmax response (solid black), with coreset (dashed green) and the long-tail algorithm (dotted red). Here again, winners are clearly identified. It is striking that the first 25,000 was that were passively sampled by "long-tail" , achieved similar accuracy to 10000 points actively queried by active-iNAS in CIFAR-10 (and less than 15000 points for CIFAR-100). Concluding that the active-iNAS achieves better accuracy for any budget compared to the long tail setting.

|     |     |
|:---:|:---:|
| (a) CIFAR-10 | (b) CIFAR-100 |

Figure 2: Comparison of active-iNAS to coreset of Geifman and El-Yaniv [9] across two datasets, (a) CIFAR-10, (b) CIFAR-100. In black (solid) – active-iNAS over softmax response, green (dashed) – active-iNAS over coresets and red (dotted) – coreset (Geifman and El-Yaniv).

## Footnotes

[1]SVHN contains 73,256 instances and was, therefore, trimmed to 50000.