[Reviews · NeurIPS 2019]

Reviewer 1



This paper provides an active learning approach that targets both the selection of examples and the most suitable architecture to learn an image recognition classifier. The model searches for effective architectures while applying active learning. The authors apply three query techniques: softmax response, MC-dropout, and coresets. The authors show that the model outperforms a traditional active learning approach based on static architecture. The paper is well-written and presented. The general idea of architecture learning is clearly not novel but what really matters is an instantiation of the idea. The authors present a concrete approach, which is reasonable and supported by some theoretical considerations. However, I have some doubts on the generality of the results. The baseline architecture is included in the set of networks the model can generate this means that: (i) the approach cannot be claimed to improve a standard model for image classification as there are more powerful architectures; and (ii) the paper just shows that the proposed approach can select a richer architecture than the baseline.

Reviewer 2



This paper proposed a method for doing active learning (AL) where in each AL iteration the optimization is done over network architecture and the underlying parameters, as opposed to other methods which fixes the architecture and only optimizes the parameters. These two optimizations are done separately, by first performing a local search among models of monotonically increasing complexity and then optimizing parameters of the obtained architecture. The authors used this method with three different active learning algorithms and showed that their method improved performance of these ALs. The paper is very well-written and clear. The problem of architectural optimization is also of great importance in the field. One confusing thing is that the title and introduction of the paper sound as if the paper is an AL paper, whereas it is mostly addressing architecture optimization in an AL framework (which can be replaced by any querying function, including passive random selection). Therefore, it is expected that the paper gives a more comprehensive review about existing methods of architectural search (rather than AL methods). The supplementary material does include a review of the existing architecture search methods, some of which should be moved to the main text. It is not clear why a new network architecture search is needed, and what makes it more suitable for AL settings. Any existing NAS method, including those based on reinforcement learning, Bayesian optimization, etc, could be used within AL iterations, why should iNAS be a better choice for this purpose? Moreover, is the proposed algorithm scalable to large data sets as it has to train/test multiple model architectures in each AL iterations? Does the model get fine-tuned in each iNAS iteration, or it gets trained from scratch? In the early AL iteration when there is small number of labeled samples, how reliable is the architecture selection as the size of the validation fold would be very small? Finally, it might not always be the case that more data induce deeper (more complicated) model. One can even imagine a scenario when smaller number of samples might lead to selecting a model with higher complexity as the underlying pattern in the data distribution might not be visible when there is smaller number of samples. Why did the authors allow the algorithm to only move towards increasing complexity in each iNAS iteration?

Reviewer 3



The main novelty of this work lies in the proposed algorithm for selection of an optimal network architecture, given the pre-defined AL selection criterion. The advantage of this is that this approach can select on-the-fly a potentially less complex model under the same budget, while also attaining improved performance in the classification task. The authors also provide theorethical motivation for this approach using the notion of tight bounds for the VC-dimension of deep neural nets. Experimental validation on three benchmark datasets show improvements (sometimes marginal) over the fixed baseline architectures. The approach is well presented and the paper is easy to read.

[Author Response · NeurIPS 2019]

We thank the reviewers for the thoughtful comments. We will address ALL comments in our final version.

**Reviewer 1:** We do not claim that we improve a standard model for image classification. We claim that in the context of active learning, the proposed method achieves enhanced speedup rates (budget/accuracy trade-off) compared to working with a fixed architecture along the active learning process. We also state that selecting the baseline architecture requires hindsight knowledge over the learning problem at hand, that might not be available when actively learning on new problems and domains. Thus, instead of selecting a fixed architecture (based on an educated guess) we recommend to build an architecture search space (similar to the one proposed for resnet), and run active iNAS. This will allow to actively learn faster with less labels and reach better results compared to the (guessed) fixed architecture. Of course, for any particular dataset it's likely that the SOTA passive architecture achieves better final result than our final architecture. However, our sample complexity speedup performance is most likely to be superior. And once you have a full labeled dataset you can take the SOTA architecture anyway. For better understanding of this point please see comparison to other active learning papers described in the supplementary material.

**Reviewer 2:** AL vs NAS - The paper discusses an enhanced active learning method that incorporate a NAS algorithm that is designed to support the active learning process. We will move some of the literature survey of NAS to the main text as you proposed. The active i-NAS (compared to standard NAS methods) leverages the properties of active learning process, where the training set starts small and grows along the process. Our algorithm deliberately designed this way, and in particular we avoid application of a full NAS on each active learning round which is both statistically prone to overfitting and computationally infeasible.

Weights sharing across active rounds - We train the network from scratch on each active learning round. A future research direction would be to integrate weight sharing ideas such as those propose in ENAS [1].

how reliable is the architecture selection in the early stage of AL - The variance of early stage active learning is known to be high due to variability in seed initialization. In our case we can see that the performance is relatively stable among repetitions (very small shadowed standard error in the curves). In most cases, Active iNAS reached the same architectures (both on the early and later stage of the active learning). The latter result is not presented in the paper and will be added.

It might not always be the case that more data induce deeper (more complicated) model ... - We think that this situation is statistically rare in i.i.d selection of the initial seed, and does not happen in the later stages of active learning where points to be labeled are selected to increase diversity and improve model accuracy on uncertain cases.

**Reviewer 3:** Large budget compared to classic AL - typical deep neural models have huge hypothesis class capacity, which make them hungry for labeled training sets. In this setting it does not make sense to use the budgets that have been used in classic AL (5-300). Please see the references to deep active learning papers as it is a common numbers/sizes for deep learning AL.

Insights about why your approach remains better compared to the fixed networks - the improvement comes from both being more adapted to the training set size, and due to enhanced querying. The improvement in querying performance, is demonstrated in Section 4.4. In the experimental results, we can see that the initial architecture selected by iNAS outperforms the (large) fixed architecture in the early active learning stage in most cases. This supports the claim that the architectures that have been selected are more suitable to the training set size along the active learning process. We will emphasize this point in the text.

Lack of comparisons with other state-of-the-art approaches/architectures for AL - We compared all methods for active learning (Coreset, softmax and MC-dropout), on our experimental setting and architectures. Please see a direct comparison to the state-of-the-art papers for deep active learning in the supplementary material. In addition, please see the response for reviewer 1 about state of the art architecture selection in hindsight.

Random querying baseline – we agree that random querying (i.e. passive learning) is a standard baseline that any good AL algorithm should surpass. Our algorithm easily beats passive learning in all cases. We will add the learning curves. You can also see some evidence that we are better than passive learning by considering the results of our other baselines (in their original papers). For example, Sener & Savarese and Geifman & El-Yaniv.

Comparison to cross-validation over several fixed architectures - The active iNAS allows us to actively (wisely) search in a large architectural search space (60 architectures in our case). Searching exhaustively in this search space is computational in-feasible, and prone to over-fitting (due to the small validation set in early stages). Thus, we proposed the iNAS algorithm that leverages the incremental expansion of the training set size in active learning and expand the selected architecture accordingly in much efficient way compared to exhaustive cross-validation.

[1] Pham, Hieu, et al. "Efficient neural architecture search via parameter sharing." arXiv preprint arXiv:1802.03268 (2018).

[Meta-Review · NeurIPS 2019]

This paper proposes a strategy for an efficient deep network architecture search (here for image classification, but a the general idea would apply for other tasks as well). The proposed strategy is will motivated and involves a data sampling stage at each step. Here, an active querying strategy can be employed and the authors evaluate their strategy with three different active sampling strategies. They show that their strategy improves over active learning (with the same active query strategies) with a fixed architecture. However, the reviewers have rightly pointed out that a comparison with other architecture search strategies would also have been in place. The work in this submission is solid, and may become a useful reference for other research groups or practitioners. (As a side comment: "Deep Neural Architecture Search with Active Learning" may have been a more suitable title.)